# Diverse Epidemiology of *Leptospira* Serovars Notified in New Zealand, 1999–2017

**DOI:** 10.3390/pathogens9100841

**Published:** 2020-10-14

**Authors:** Shahista Nisa, David A. Wilkinson, Olivia Angelin-Bonnet, Shevaun Paine, Karen Cullen, Jackie Wight, Michael G. Baker, Jackie Benschop

**Affiliations:** 1Molecular Epidemiology and Public Health Laboratory, Hopkirk Research Institute, School of Veterinary Science, Massey University, Palmerston North 4474, New Zealand; s.nisa@massey.ac.nz; 2New Zealand Food Safety Science and Research Centre, Massey University, Palmerston North 4474, New Zealand; david.wilkinson@univ-reunion.fr; 3School of Fundamental Sciences, College of Sciences, Massey University, Palmerston North 4474, New Zealand; Angelin-bonnet@massey.ac.nz; 4Institute of Environmental Science and Research, Wellington 5022, New Zealand; Shevaun.Paine@esr.cri.nz (S.P.); karen.cullen@esr.cri.nz (K.C.); Jackie.Wright@esr.cri.nz (J.W.); 5Department of Public Health, University of Otago, Wellington 6021, New Zealand; michael.baker@otago.ac.nz; 6Global Leptospirosis Environmental Action Network, World Health Organization, Geneva, Switzerland

**Keywords:** leptospirosis, serovars, New Zealand, epidemiology

## Abstract

Leptospirosis in New Zealand has been strongly associated with animal-contact occupations and with serovars Hardjo and Pomona. However, recent data suggest changes in these patterns, hence, serovar-specific epidemiology of leptospirosis from 1999 to 2017 was investigated. The 19-year average annual incidence is 2.01/100,000. Early (1999–2007) and late (2008–2017) study period comparisons showed a significant increase in notifications with serovar Ballum (IRR: 1.59, 95% CI: 1.22–2.09) in all cases and serovar Tarassovi (IRR: 1.75, 95% CI: 1.13–2.78) in Europeans and a decrease in notifications with serovars Hardjo and Pomona in all cases. Incidences of Ballum peaked in winter, Hardjo peaked in spring and Tarassovi peaked in summer. Incidence was highest in Māori (2.24/100,000) with dominant serovars being Hardjo and Pomona. Stratification by occupation showed meat workers had the highest incidence of Hardjo (57.29/100,000) and Pomona (45.32/100,000), farmers had the highest incidence of Ballum (11.09/100,000) and dairy farmers had the highest incidence of Tarassovi (12.59/100,000). Spatial analysis showed predominance of Hardjo and Pomona in Hawke’s Bay, Ballum in West Coast and Northland and Tarassovi in Waikato, Taranaki and Northland. This study highlights the serovar-specific heterogeneity of human leptospirosis in New Zealand that should be considered when developing control and prevention strategies.

## 1. Introduction

Leptospirosis is a neglected zoonotic disease that can cause mild to severe febrile illness, renal and hepatic failure, and death [1]. It is estimated to cause 1.03 million cases and 58,900 deaths annually [2] with a loss of 2.90 million Disability Adjusted Life Years (DALYs) [3]. However, because leptospirosis presents as an undifferentiated febrile illness, the global burden is likely underestimated. It can be misdiagnosed as influenza, malaria or dengue fever, especially in tropical settings, testing for leptospirosis is difficult and many countries do not have a notification system [4]. Adding to the underestimation of burden, approximately 30% of patients have symptoms that persist for several years [5,6].

The disease is caused by spirochetes of the genus *Leptospira*. Until the early 2000s, there were fewer than 20 known species of *Leptospira* [7]. With the advancement of whole genome sequencing, known species increased to 64 by 2019 [8,9]. To date, there are over 300 serovars associated with these species; however, the genetic and serological classification show poor correlation [8]. Current genetic tests cannot be used to identify serovars and the most widely used test for diagnosing *Leptospira* serovars, i.e., the microscopic agglutination test (MAT), cannot be used to type species; thus, it is difficult to fully characterize the etiological agent with any one test [1]. In addition, since one or several species can belong to the same serogroup, albeit as different serovars, MAT assays can also have cross-reactivity and lead to misclassification in countries with many circulating serovars. Pathogenic species of *Leptospira* colonize the kidneys of mammals, including livestock, rodents, pinnipeds and bats, as well as birds, amphibian and reptiles [10]. Renal colonization leads to intermittent urinary shedding and infection is transmitted either through direct contact with infected urine or indirectly via the contaminated environment [4]. High-risk groups vary by location and include people that are in contact with livestock, e.g., farmers and meat workers, people in contact with rodents, e.g., sewage workers, rice paddy workers, subsistence farmers and urban slum dwellers, people involved in water-based recreational activities and people affected by extreme weather events, such as flooding [4]. The burden of disease is often higher in tropical regions with incidence greater than 10/100,000 compared to temperate countries where incidences range between 0.1–10/100,000 [11]. A systematic review of studies from 1970 to 2008 revealed highest estimates for leptospirosis morbidity and mortality in Oceania, the Caribbean, Southeast Asia, East Sub-Saharan Africa and Andean, Central, and Tropical Latin America [2].

In New Zealand (part of Oceania), leptospirosis was first identified in both humans and animals in the 1950s [12]. Incidence peaked in 1971 to 30/100,000 with dairy farmers, meat workers and pig farmers identified as having high-risk occupations [13]. At that time, cattle were the recognized maintenance host for serovars Hardjo and Pomona and pigs for serovars Pomona and Tarassovi [13]. In the early 1980s, dairy cattle and pig vaccination programmes against serovars Hardjo and Pomona were implemented [14,15]. By 1998, the annual incidence in humans declined to 3.0/100,000 [16] and by 2015 the incidence had further declined to 1.4/100,000 [17]. Analyses of New Zealand’s notification data by Thornley et al. between 1990–1998 [16] and El-Tras et al. between 2010–2015 [17] showed that incidence remained highest in meat workers and farmers. Serological surveys in New Zealand meat workers conducted from 2008 to 2011 revealed risk of infection with serovars Hardjo and Pomona in sheep abattoirs [18]. Case studies of outbreaks that occurred in 2010 and 2015 demonstrate that Hardjo and Pomona disease occurs in dairy farm workers on farms with no or incomplete vaccination programmes [5,19]. Thus, despite Hardjo and Pomona vaccine availability, both serovars still pose a risk to human health. Thornley et al. also described an increasing incidence of Ballum between 1990–1998, a serovar maintained by rodents, as the third most prevalent serovar after Hardjo and Pomona [16]. Between 2010–2015, the number of Ballum cases exceeded the number of Pomona cases, becoming the second most prevalent serovar in New Zealand [17].

The previous studies suggest changes in serovars associated with leptospirosis in New Zealand. To better understand these notification data, serovar-specific epidemiology over 19 years (1999–2017) was explored. This work describes the total and serovar-specific yearly and monthly trends, demography, hospitalization, and geographical locations. The findings suggest that there is different epidemiology associated with the different serovars in New Zealand which will require different intervention and control strategies.

## 2. Results

### 2.1. Total and Serovar-Specific Incidences

The study population comprises 1520 confirmed and 107 probable cases (*n* = 1627) of human leptospirosis notified in New Zealand from 1 January 1999 to 31 December 2017. The average annual incidence for this 19-year period is 2.01/100,000 population. The national incidence is significantly lower in the late study period (2008–2017 = 1.63/100,000) compared to the early study period (1999–2007 = 2.48/100,000) with an IRR of 0.65 (95% CI: 0.59–0.72, Table 1). Loess smoothed time series show that this decrease was not linear; there was increased incidences in the early 2000s followed by a steady decline until 2014, after which incidences steadily increased until 2017 to 2.89/100,000 (Figure 1A).

Serovar data were available for 72.4% (1178/1627) of the study population. Nine serovars were recorded in this 19-year period, with four serovars dominating notifications, i.e., Hardjo at 41.8% (492/1178), Pomona at 22.4% (264/1178), Ballum at 21.4% (253/1178) and Tarassovi at 9.3% (109/1178), while serovars Copenhageni (*n* = 35), Canicola (*n* = 10), Australis (*n* = 10), Grippotyphosa (*n* = 4) and Bratislava (*n* = 1) combined accounted for 5.1% (60/1178) of cases. 

Serovar-specific yearly incidence for Ballum increased significantly in the late study period (IRR: 1.59, 95% CI: 1.22–2.09, Table 1). The incidence of Tarassovi and cases with unknown serovars also increased over the study period; however, this was not significant (Table 1). Hardjo and Pomona incidences decreased significantly between the early and late study period, with an IRR of 0.5 for both serovars (Table 1). Hardjo and Pomona yearly trends mirrored the yearly trends of all cases (Figure 1B).

The average monthly incidence for all cases during the study period remained approximately the same (0.17/100,000, Figure 2A). However, serovar-specific average monthly incidences showed a May–July (winter) peak with Ballum (0.04/100,000), an August–November (spring) peak with Hardjo (0.07/100,000) and a November–March (summer) peak with Tarassovi (0.02/100,000, Figure 2B).

#### 2.1.1. Sex and Age

Data on sex were available for 99.4% (1617/1627) of all cases and age was available for all except one case. Males accounted for 89.6% (1449/1617) of cases with an average annual incidence of 3.65/100,000. Females accounted for 10.4% (168/1617) of cases with an average annual incidence of 0.41/100,000. Median ages of males (43 years, IQR 32–52), and females (43 years, IQR 30.5–52) were the same (Figure 3A) and the distribution of ages shows 40–49 years with the highest number of notifications for both sexes (Figure 3B). Serovar-specific median age shows Ballum cases were older (males: 49 years, IQR 39.5–56, and females: 51 years, IQR 43.25–55.75), while Tarassovi cases were younger (males: 39 years, IQR 28–49, females: 38 years, IQR 27.5–44, Figure 3C). The most notified age group for Ballum was 50–59 years, for Hardjo and Pomona it was 40–49 years and for Tarassovi it was 20–29 years (Figure 3D).

#### 2.1.2. Ethnicity

Ethnicity data were available for 92.8% (1509/1627) of all cases. Europeans accounted for 79.3% (1197/1509) of cases with an average annual incidence of 2.15/100,000 and Māori accounted for 18.2% (275/1509) of cases with an average annual incidence of 2.24/100,000. Other ethnicities notified included Pacific peoples (*n* = 26), Asians (*n* = 10) and other (*n* = 1). Loess-smoothed time series plots show that the total yearly incidence of Europeans cases remained similar throughout the study period, whereas the incidence of Māori cases followed the total incidence trend, i.e., an increase in the early 2000s followed by a steady decline, before increasing in 2017 (Figure 4). Loess-smoothed time series plots of ethnicity incidences stratified by serovar demonstrate only Hardjo having a similar trend in Europeans and Māori (Figure 4). Overall, the incidence of Ballum and Tarassovi was higher in Europeans, incidence of Pomona was higher in Māori and the incidence of Hardjo was similar in both ethnic groups (Table 1). Comparison between early and late study periods showed significant increase of Ballum cases in both ethnic groups, significant decrease of Hardjo and Pomona cases in both ethnic groups and significant increase of Tarassovi cases in Europeans (Table 1).

#### 2.1.3. Occupation 

Occupation data were recorded in 93.6% (1523/1627) of all cases. Meat workers represented 31.6% (481/1523) of cases with an average annual incidence of 140.61/100,000. Stratification by serovar showed Hardjo (57.59/100,000) and Pomona (45.31/100,000) predominated in meat workers. Comparison between the early and late study periods showed a significant decrease in serovars Hardjo and Pomona in meat workers (Table 1). Farmers represented 33.2% (506/1523) of cases with an average annual incidence of 57.83/100,000 (Table 1). Stratification by serovar showed Hardjo (18.74/100,000) and Ballum (11.09/100,000) predominated in farmers. Comparison between the early and late study periods showed a significant increase in cases with Ballum and unknown serovars amongst farmers. Dairy farmers represented 14.2% (216/1523) of cases with an average annual incidence of 43.89/100,000. Stratification by serovar showed Hardjo (12.80/100,000) and Tarassovi (12.59/100,000) predominated in dairy farmers. Comparison between the early and late study periods showed a significant increase in cases with Ballum and unknown serovars amongst dairy farmers. Other occupations represented 26% (424/1627) of all cases with an average annual incidence of 0.53/100,000. Stratification by serovar showed that Ballum (0.16/100,000) and unknown serovar (0.22/100,000) predominated in other occupations. Comparison between the early and late study periods showed a significant increase in cases with Ballum and unknown serovars and a decrease in Pomona cases in other occupations (Table 1).

#### 2.1.4. Hospitalization

Hospitalization data were available for 84.5% (1375/1627) of all cases where 53.8% (740/1375) were hospitalized. The rate ratio of hospitalized and non-hospitalized cases for all cases and any serovar shows leptospirosis cases were more likely to be hospitalized than not (RR: 1.16, 95% CI: 1.05–1.29). When the data were stratified by ethnicity, there were no significant differences between hospitalized and non-hospitalized cases in either Europeans or Māori. However, stratification of hospitalization data by serovar shows that European cases with Ballum and Pomona are significantly more likely to be hospitalized than Māori cases with these serovars (Table 2). There was one fatality in the study population, however, it is unknown if the death was due to leptospirosis.

#### 2.1.5. Spatial Pattern

The highest average annual incidence for all cases was in West Coast (7.84/100,000), Hawke’s Bay (7.03/100,000) and Tairawhiti (6.91/100,000, Figure 5). This pattern was similar when ethnicity was stratified by European and Māori with the exception of Māori having a much higher incidence in Hawke’s Bay (11.56/100,000), Tairawhiti (9.59/100,000) and Whanganui (9.01/100,000) than Europeans (Figure 5).

Comparison of incidences between the early and late study periods showed that period average increased in Whanganui and decreased in South Canterbury for all cases (Appendix A). This pattern was broadly similar when period average incidence was stratified by ethnicity (European and Māori), except for Waikato where there was an increase in European leptospirosis incidence from the early to the late period and a decrease in Māori incidence. The overall decrease in South Canterbury was due to a large decrease in incidence of Europeans (Appendix A).

District Health Board incidences stratified by serovar showed Ballum predominated in West Coast and Northland in all cases and Europeans while for Māori, Ballum predominated in West Coast and Wairarapa (Figure 6). For the remaining three serovars, Hardjo, Pomona and Tarassovi, the spatial patterns were similar for all cases and for cases stratified by ethnicity, i.e., Hardjo predominated in Hawke’s Bay, Pomona predominated in Hawke’s Bay and Tairawhiti, and Tarassovi predominated in Waikato, Taranaki and Northland (Figure 6).

## 3. Discussion

This is the first study that has investigated the heterogeneity of leptospirosis associated with the different serovars. The data show that while the overall incidence of leptospirosis in New Zealand is decreasing, serovar-specific incidence displays different trends that can inform more targeted prevention and control programmes. 

The 19-year (1999–2017) average annual incidence of leptospirosis in New Zealand is 2.01/100,000, a significant decline from the previous assessment between 1990–1998 (4.4/100,000, IRR: 0.46, 95% CI: 0.43–0.49) [16]. The overall demography of cases has remained the same as the previous assessment where 90% of cases are males between 20–60 years of age. This may be due to high-risk occupations being male-dominated as women are less likely to work in meat-processing activities [20], however, many women engage in farming in New Zealand [21]. Overall, incidences by ethnicity were similar and occupational incidence was high in meat workers (Table 1). 

Temporal analysis shows a complex trend with a year-on-year increase in the last 3 years of the study (Figure 1A). Serovar-specific analysis shows that the increase in the last 3 years appears to be largely associated with an increase in cases that did not have serovar data, i.e., serovar unknown (Figure 1B). This is likely due to the use of PCR as a diagnostic test since 2013 (Appendix A), which only identifies a case as either being positive or negative for *Leptospira* [22]. Further serological and genetic analysis is required to discern what constitutes the unknown serovars in New Zealand.

The overall decrease in incidence is reflective of the two dominant serovars, Hardjo and Pomona. The continued decline of serovars Hardjo and Pomona since the implementation of dairy cattle and pig vaccine programs for these serovars suggests that vaccination remains an effective measure in reducing human cases. However, consistent with the previous studies, Hardjo and Pomona still disproportionally burden meat workers (Table 1) [16,17]. This may be attributed to the fact that while 99% of dairy cattle herds are vaccinated [23], vaccination of dry stock is less frequent, i.e., beef cattle herds (18–25%), deer herds (5–9%) and sheep flocks (<1%) [24]. Concerningly, the roles meat workers perform place them at almost continuous risk of urine exposure from dry stock, e.g., yarding, stunning, skinning, and gutting [18]. Although wearing of personal protective equipment (PPE) in the meat industry is mandated, there are issues with compliance and the logistics of its use. Dreyfus et al. reported that wearing gloves, facemasks and goggles were not protective factors in multivariable model investigating risk of exposure in sheep meat workers in New Zealand abattoirs [25]. The geographical distribution and stratification by occupation and ethnicity in this work has revealed that Māori meat workers, especially in Hawke’s Bay and Tairawhiti, are infected with Hardjo and Pomona, likely from unvaccinated stock (Figure 6). Interestingly, while Hardjo and Pomona incidences are similar in meat workers, Hardjo incidences are twice those of Pomona in farmers and 12 times more in dairy farmers (Table 1). One could hypothesize that this, in part, could be due to serovar Balcanica. Balcanica is in the same serogroup as Hardjo, thus it reacts serologically to Hardjo. Balcanica has been found in New Zealand cattle and brushtail possums (Trichosurus vulpecula) and possums are believed to be the maintenance host for Balcanica with cattle as a bridge host [26,27]. As possums forage on pastures, farmers are likely to be exposed to possum urine, thus, Balcanica infection in farmers may be recorded as Hardjo. Monthly analysis shows Hardjo has a high incidence between August–November (spring, Figure 2B). This period coincides with lambing, beef cattle calving and calving on most New Zealand dairy farms. Assisting with calving has been shown to be a risk factor for *Leptospira* seropositivity in New Zealand farmers, probably through direct exposure to infected animals whilst assisting with birthing [24]. Exposure is also likely to increase during spring with the introduction of both new heifers and new workers to the dairy farm environment. 

While Hardjo and Pomona incidences decreased, the incidence of Ballum significantly increased over the study period in all ethnicities and in all occupations except meat workers. Rodents are the typical maintenance hosts for Ballum worldwide [28] and a cross-sectional study of cattle and wildlife on a New Zealand dairy farm showed Ballum seroprevalence in cattle (12.3%), *Mus musculus* (mice: 31.6%), *Rattus rattus* (rats: 25%) and *Erinaceus europaeus* (hedgehogs: 58.3%) [29]. Thus, Ballum may be maintained in New Zealand wildlife and contact with wildlife urine, e.g., handling rodent-contaminated animal feed, working in pest control and the general population dealing with rodents in their homes, may pose a risk. Unlike the other serovars, which were low in non-occupational groups, approximately 49% of Ballum cases were from other occupation groups. Average monthly incidences of Ballum were highest between May–July (autumn-winter, Figure 2). This peak in Ballum may indicate rodents moving into warmer places such as houses and sheds. 

The incidence of Tarassovi increased significantly in Europeans, with dairy farmers having a relatively high incidence compared to the other occupational groups (Table 1). Monthly incidences were low over winter and geographical distribution showed high incidence of Tarassovi in Northland, Waikato and Taranaki. These regions are known for intensive dairy farming in New Zealand. In addition, New Zealand’s milk production is highly seasonal, thus a winter trough is likely a reflection of the dairy productivity cycle, where most dairy herds (spring calving herds) are not milked between May–July, thus decreasing dairy farmers’ direct exposure to cattle urine over winter. A 2016 cross-sectional study of 200 dairy farms in New Zealand showed high shedding prevalence and serological evidence of Tarassovi in Northland and Waikato [23]. Thus, dairy cattle appear to be a major risk factor for Tarassovi infections in New Zealand. 

This work also looked at hospitalization of cases. Cases with leptospirosis were significantly more likely to be hospitalized than not but stratification by ethnicity and serovar shows only European cases were more likely to be hospitalized and only when they were infected with Ballum or Pomona (Table 2). Further work with multivariable analysis, including adjusting for the confounding effects of age and sex will be applied to evaluate this disparity in the hospitalization rates ratios between Europeans and Māori. 

The overall patterns of leptospirosis in New Zealand varied when compared to other temperate countries. Like New Zealand, improved diagnostics saw an increase in cases in Ireland with high-risk activities associated with occupations involving livestock [30]. However, unlike New Zealand, exposure in Ireland was also highly associated with recreational activities, whereas exposure in Germany was as likely to be occupation as recreational [31], while in Italy exposure was more likely to be recreational than occupational [32]. Interestingly, unlike New Zealand where the total monthly incidence of leptospirosis showed little change during the year, the seasonal pattern in European countries including Germany, Italy, France and Slovakia, showed high incidences between July to October [30,31,32,33,34]. While this observation is interesting, results from this study have shown that the total monthly incidence is not a good way to determine leptospirosis seasonality in New Zealand as each serovar has a different monthly pattern, likely due to different exposures and activities, e.g., Hardjo and Pomona in meat workers and Tarassovi in dairy farmers.

It is important to note that while this study described total and serovar-specific trends, there are potential biases with this analysis. For example, the change in diagnostics can lead to misclassification of cases. If a region with a high incidence of Hardjo or Pomona switched from serological to PCR testing, this analysis would identify it as a decrease in Hardjo and Pomona cases in the late study period since the PCR positive Hardjo and Pomona cases would be classified as unknown serovar. In addition, the occupational incidence of farmers may be over-estimated because the population at risk denominator used the employed census of usually resident population. This denominator would not include migrant/seasonal/temporary workers and family/friends who may help at the farm but who may not have a farming occupation. Lastly, there are differences in trends seen in Europeans and Māori that may be attributed to physiological response to disease, lifestyle, health-seeking behaviors and changes in the at-risk population over time, e.g., the number of meat workers decreased by 23% from the early to the late study period. 

### 3.1. Implications for Surveillance and Research

Reporting the type of farming should be improved within the current surveillance framework. Currently, 87% (440/506) of farmers did not classify their type of farming. This information would provide further insight into at-risk groups and thus intervention strategies. 

While the use of PCR provides rapid diagnostics, it has also led to loss of serovar data, thus greatly reducing the utility of the surveillance system to inform prevention and control strategies [35]. Research should look at improving molecular diagnostics to incorporate serovar discrimination. 

### 3.2. Implications for Prevention and Control

Due to the varying epidemiology exhibited by the different serovars, mitigation and control will not be uniform. 

Vaccination of dry stock should be considered as meat workers are still exposed to serovars in current vaccine, as only the pig [15] and dairy cattle industries [23] are approaching full coverage. 

Critical research needs to include developing animal vaccines for serovars that are in livestock if there is a clear identification that the livestock is the source of human infection, i.e., Tarassovi from dairy cattle.

Regionally targeted control strategies should be considered in consultation with key stakeholders and affected industries, e.g., rodent control in regions with high Ballum cases. While this may sound ambitious, New Zealand has a Predator Free 2050 plan put forward by the government and rat eradication has been achieved on smaller islands [36]. 

Implementation of occupational health strategies should include increased awareness of risk factors, improving hygiene and improving the use of PPE to help reduce exposure when dealing with unvaccinated animals. 

Awareness should be raised in regions with historically low incidence of leptospirosis that have undergone a significant shift in farming practice (intensive dairy farming in Canterbury and Southland) [37]. 

Recreation is a significant risk factor for leptospirosis in developed countries [38], thus, increasing awareness in these communities, i.e., exposure to wildlife via hunting and water sports, adventure sports and travel, is important. 

## 4. Materials and Methods 

Data source: Routinely collected leptospirosis surveillance data from 1 January 1999 to 31 December 2017 were extracted from New Zealand’s notifiable disease database (EpiSurv) maintained by the Institute of Environmental Science & Research Ltd. (ESR) on behalf of the Ministry of Health. This study utilized data collected on cases status, report date, sex, age, ethnicity, occupation, exposure to animal species at place of employment, hospitalization, death, laboratory tests, infecting serovar, and location of cases at District Health Board (DHB) level. 

Inclusion criteria and case definitions: Leptospirosis cases were included in the analysis if they had a case status recorded as confirmed or probable [39]. Cases under investigation or with unknown status were excluded from the analysis. Confirmed cases were defined as having a clinically compatible illness, in addition to laboratory definitive evidence which included at least one of the following: (a) isolation of *Leptospira* spp. from clinical specimen, (b) detection of *Leptospira* DNA by polymerase chain reaction (PCR), (c) a single antibody titre of ≥400 in MAT, or (d) a four-fold or greater rise in MAT titre between acute and convalescent sera [39]. Probable cases were defined as having a clinically compatible illness and laboratory suggestive evidence of a single raised MAT titre of <400. The current panel for MATs in New Zealand includes eight serovars: Australis, Ballum, Canicola, Copenhageni, Grippotyphosa, Hardjo, Pomona and Tarassovi. Serovar Bratislava data were available up until 2002, after which its surveillance was discontinued. All 9 serovars were included in the analysis. A total of 1695 cases of leptospirosis were reported in the study period, of which 1627 met the inclusion criteria. Due to the small number of probable cases, cases were not stratified by confirmed and probable. All analysis was performed on total cases. 

Demography: Demographic data included sex, age, ethnicity and occupation. Age was in years and examined by median age with interquartile ranges (IQR) and with 10-year age breaks. Ethnicity data used the New Zealand convention of prioritized ethnicity. Where multiple ethnicities were recorded, Māori ethnicity took precedence, followed by Pacific peoples, Asians, then Europeans, with all remaining ethnicities classified as other [40]. Temporal, spatial and serovar analyses by ethnicity were only performed if there were over 50 cases per ethnicity. Occupations recorded in the EpiSurv database were grouped into 4 broad categories: meat worker, farmer, dairy farmer and other occupation. Meat worker included all slaughterers, boners, slicers and general abattoir workers. Farmer included dry stock, mixed and farmers that did not classify the type of farming. Dairy farmer included all dairy farmers, milkers or a farmer/farm worker whose exposure to animal species at place of employment included only dairy cattle. All occupations that were neither a livestock farmer or a meat worker were categorized as other occupation which included crop/vegetable farmers/workers, forestry workers, office workers, as well as retired, unemployed and unknown occupations. 

Population data: Population at risk denominators for each year were interpolated by Environmental Science & Research Ltd. using the 2001, 2006 and 2013 usually resident census population data at national and District Health Board levels. These data were used to calculate incidence stratified by time, serovar, ethnicity and location. Temporal trends were visualized by report year (1999 to 2017) and report month (January to December) with locally weighted scatterplot smoothing (LOESS), a linear regression analysis tool that creates a smooth line through time plots to foresee trends. Average incidences were calculated for the entire study period (1999–2017), the early period (1999–2007) and the late period (2008–2017) using the interpolated denominators for the respective years. The population of farmers and meat workers was calculated from the list of Australian and New Zealand Standard Classification of Occupations (ANZSCO) using the 2001, 2006 and 2013 employed census of usually resident population [41]. Other occupation at risk denominator used the 2001, 2006, 2013 census of resident population minus the population of farmers and meat workers for the respective years. Incidences within occupational groups for the entire study period was calculated using the average of 2001, 2006 and 2013 employed census while early period (1999–2007) used the average of 2001 and 2006 and late period (2008–2017) used 2013 employed census. Incidence Rate Ratios (IRR) for the early and late period of all cases and cases stratified by serovar, ethnicity and occupation were compared. Total and serovar-specific rates ratio of hospitalized and non-hospitalized cases was calculated for all, European and Māori cases. Analyses by serovar were only performed if there were over 50 cases per serovar during the study period.

Spatial analysis: There are 20 District Health Boards in New Zealand, 15 in the North Island and 5 in the South Island (Appendix A). Geographical boundaries of New Zealand District Health Boards were obtained from the Ministry of Health [42]. The average annual incidence by District Health Board for all, European and Māori cases are shown on chloropleth maps for the entire study period, the early period and the late period. Average serovar-specific incidences by District Health Board in all, European and Māori cases are shown in choropleth maps for the entire study period only.

Stata 14.0 was used for EpiSurv data merging and management [43]. Data manipulation and analysis was done in R [44] using the packages dplyr [45], tidyverse [46] and forcats [47] for manipulation, epiR [48] for analysis and packages ggplot2 [49], rgdal [50], gridExtra [51], png [52], ggsn [53], maps [54] and maptools packages [55] for graphs and maps. The Incidence Rate Ratios (IRR) and hospitalization rate ratios were compared based on their Poisson confidence intervals using R’s base stats package (R version 4.0.1) [39].

Ethical considerations: This study was approved by the Massey University Human Ethics Committee in November 2017 (Application ID: 4000018726). 

## 5. Conclusions

In conclusion, seasonal patterns together with spatial and demographic data reveal that the epidemiology of leptospirosis is heterogeneous in New Zealand. Incidence of serovars Hardjo and Pomona is decreasing over time and is associated with meat workers in Hawke’s Bay and Tairawhiti, incidence of Tarassovi is increasing over time and is associated with dairy farmers in Waikato, Taranaki and Northland, and incidence of Ballum is increasing over time and is associated with farmers and the general population. Thus, mitigation cannot use a one-size-fits-all strategy but may require different strategies depending on the risk groups, prevalent serovars and their maintenance host, including livestock and wild animal species.

## Figures and Tables

**Figure 1 pathogens-09-00841-f001:**
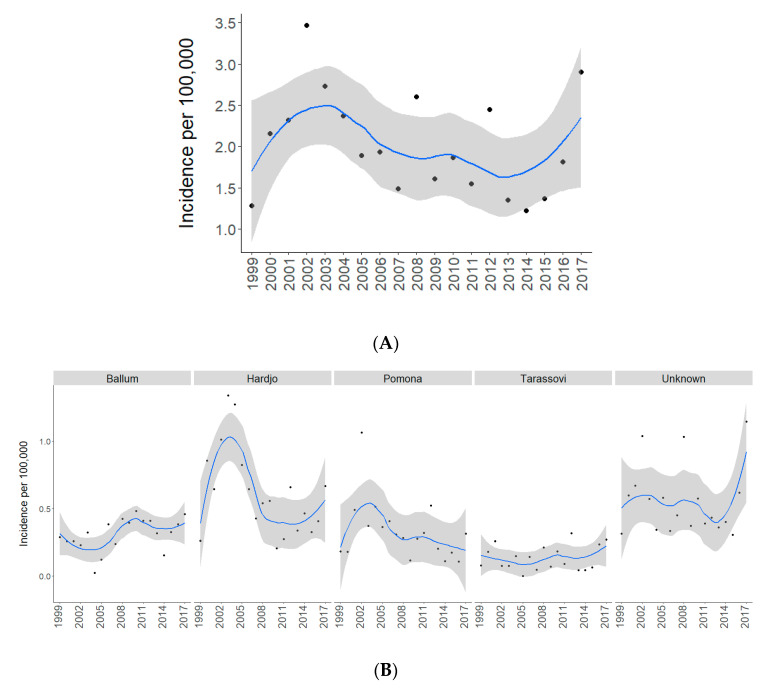
Time series of yearly incidence of notified leptospirosis cases in New Zealand, 1999 to 2017. (**A**) Total annual incidence per 100 000 and (**B**) serovar-specific annual incidence per 100,000. Dots indicate yearly incidence and blue lines represent Loess-smoothed incidence with 95% confidence intervals shown in grey.

**Figure 2 pathogens-09-00841-f002:**
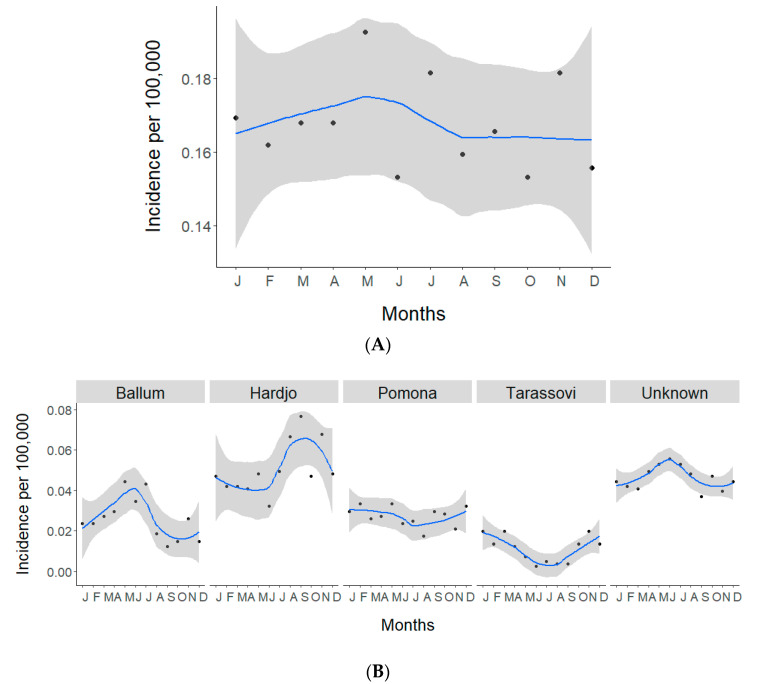
Time series of monthly incidences of notified leptospirosis cases in New Zealand, 1999 to 2017. (**A**) Total average monthly incidence per 100,000, (**B**) serovar-specific average monthly incidence per 100,000. Each dot indicates the average monthly incidence of 19 years and blue lines represent Loess-smoothed incidence with 95% confidence intervals shown in grey.

**Figure 3 pathogens-09-00841-f003:**
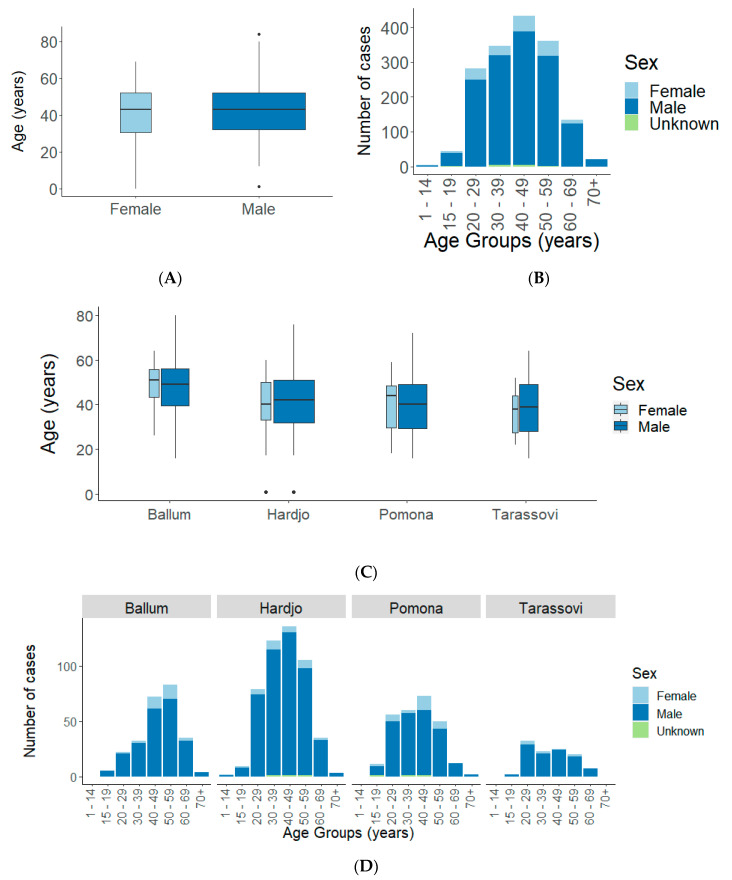
Age and sex of notified leptospirosis cases in New Zealand, 1999 to 2017. (**A**) Box plot, (**B**) frequency histogram, (**C**) box plots stratified by serovar and (**D**) frequency histograms stratified by serovar. Box plot widths are proportional to the square root of the number of cases, box plot lengths display the interquartile range of the ages, the whiskers of the box are the minimum and maximum quartile of the ages, the line on the box marks the median ages and the dots represents outliers.

**Figure 4 pathogens-09-00841-f004:**
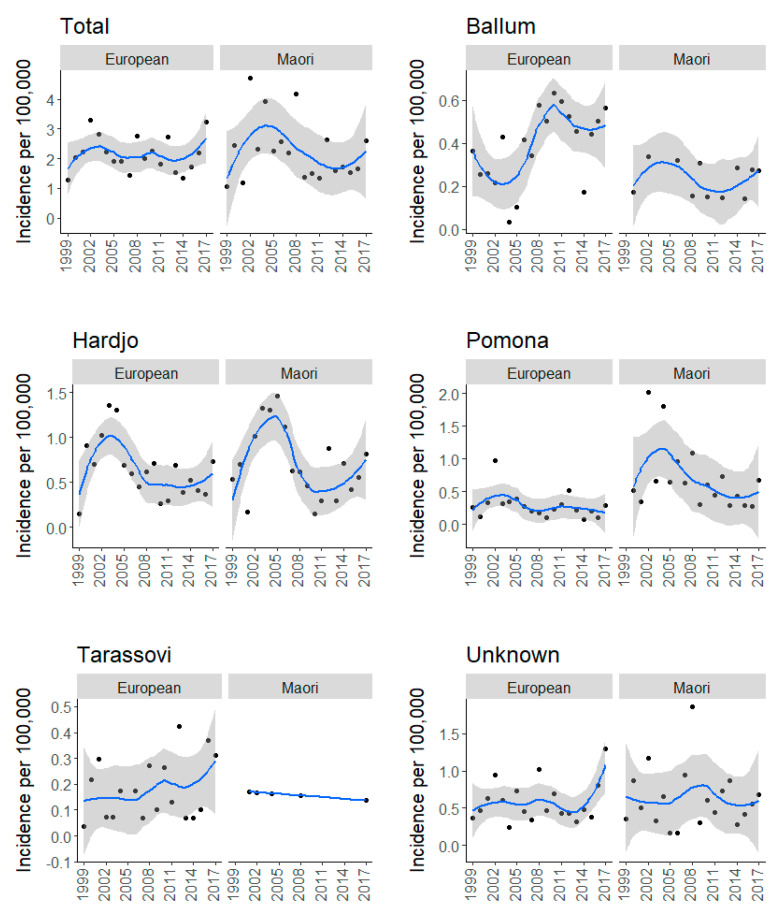
Total and serovar-specific time series of yearly incidences of notified European and Māori leptospirosis cases in New Zealand, 1999 to 2017. Dots indicate yearly incidences and blue lines represent Loess-smoothed incidence with 95% confidence intervals shown in grey.

**Figure 5 pathogens-09-00841-f005:**
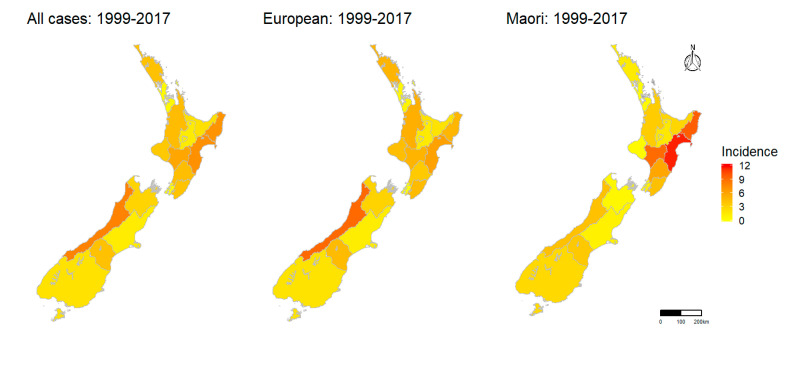
Choropleth maps of average annual incidence per 100,000 of notified leptospirosis cases by District Health Boards in New Zealand for the entire study period (1999–2017) for all cases, Europeans cases and Māori cases.

**Figure 6 pathogens-09-00841-f006:**
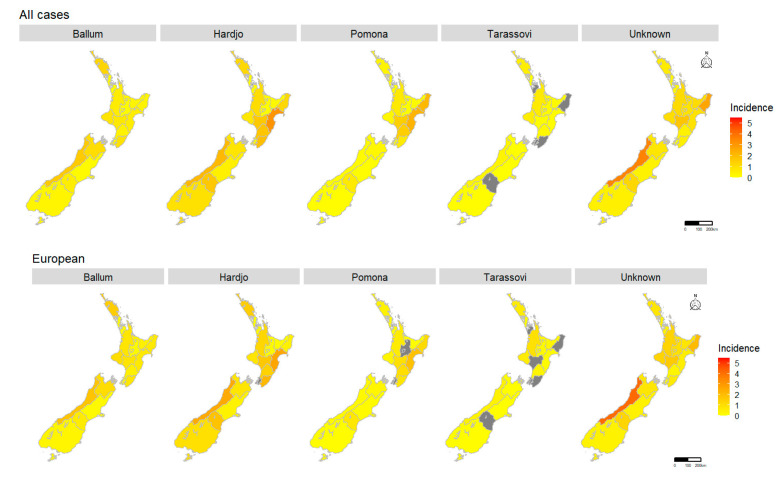
Choropleth maps of serovar-specific average annual incidence per 100,000 of notified leptospirosis cases per District Health Board in New Zealand, 1999–2017 for all cases, European cases and Māori cases. Grey indicates District Health Boards with zero incidence.

**Table 1 pathogens-09-00841-t001:** Total and serovar-specific average annual incidences of leptospirosis notifications in New Zealand for all cases and for cases stratified by ethnicity and occupation.

	Average Annual Incidence Per 100,000			
	1999–2017	1999–2007 ^a^	2008–2017 ^b^	IRR ^b/a^ (95% CI)
**All cases**				
Ballum	0.31	0.23	0.38	1.59 (1.22–2.09) ***
Hardjo	0.61	0.81	0.44	0.55 (0.46–0.67) ***
Pomona	0.33	0.43	0.24	0.56 (0.43–0.72) ***
Tarassovi	0.13	0.11	0.15	1.39 (0.83–2.11)
Unknown	0.55	0.54	0.57	1.04 (0.86–1.26)
Total	2.01	2.48	1.63	0.65 (0.59–0.72) ***
**European**				
Ballum	0.39	0.27	0.49	2.18 (1.59–2.99) ***
Hardjo	0.64	0.79	0.50	0.63 (0.51–0.78) ***
Pomona	0.28	0.36	0.22	0.62 (0.45–0.86) **
Tarassovi	0.17	0.12	0.22	1.75 (1.13–2.78) *
Unknown	0.59	0.53	0.65	1.22 (0.97–1.53)
Total	2.15	2.12	2.19	1.03 (0.92–1.16)
**Māori**				
Ballum	0.14	0.09	0.14	5.02 (1.12–46.15) *
Hardjo	0.69	0.92	0.52	0.62 (0.38–0.94) *
Pomona	0.66	0.85	0.51	0.64 (0.39–1.00) *
Tarassovi	0.04	0.06	0.03	0.56 (0.05–4.87)
Unknown	0.63	0.57	0.67	1.24 (0.77–2.02)
Total	2.24	2.53	2.00	0.79 (0.62–1.00)
**Meat worker**				
Ballum	1.75	1.71	2.00	1.30 (0.17–9.71)
Hardjo	57.29	78.60	38.67	0.55 (0.39–0.75) ***
Pomona	45.31	57.52	36.00	0.69 (0.49–0.97) **
Tarassovi	1.75	2.28	1.33	0.65 (0.06–4.54)
Unknown serovar	31.28	38.16	26.67	0.78 (0.51–1.17)
Total	140.61	180.56	109.36	0.67 (0.55–0.82) ***
**Farmer ^c^**				
Ballum	11.09	9.06	13.12	1.61 (1.05–2.49) *
Hardjo	18.74	19.55	18.23	1.04 (0.75–1.42)
Pomona	8.69	7.87	9.56	1.35 (0.84–2.19)
Tarassovi	4.00	2.86	5.11	1.98 (0.95–4.38)
Unknown serovar	14.29	14.79	14.01	1.05 (0.73–1.52) *
Total	57.83	54.85	61.37	1.24 (1.04–1.49) *
**Dairy Farmer**				
Ballum	5.59	3.04	7.53	2.75 (1.12–7.69) *
Hardjo	12.80	14.34	11.29	0.87 (0.52–1.48)
Pomona	1.02	0.87	1.13	1.44 (0.17–17.27)
Tarassovi	12.59	9.99	14.67	1.63 (0.95–2.86)
Unknown serovar	10.76	5.22	15.43	3.29 (1.69–6.87) ***
Total	43.89	33.47	52.30	1.74 (1.31–2.32) ***
**Other occupation ^d^**				
Ballum	0.16	0.11	0.21	2.14 (1.44–3.24) ***
Hardjo	0.09	0.12	0.07	0.67 (0.39–1.11)
Pomona	0.04	0.06	0.02	0.37 (0.14–0.87) *
Tarassovi	0.008	0.003	0.01	4.61 (0.52–218.12)
Unknown serovar	0.22	0.16	0.26	1.83 (1.31–2.58) ***
Total	0.53	0.48	0.63	1.46 (1.19–1.78) ***

^a^ Early period (1999–2007). ^b^ Late period (2008–2017). ^b/a^ IRR calculated as late period divided by early period. ^c^ Includes dry stock, mixed and unspecified farmers. ^d^ Includes everyone who is not a farmer or meat worker. *** *p* < 0.001, ** *p* < 0.01, * *p* < 0.05.

**Table 2 pathogens-09-00841-t002:** Total and serovar-specific rate ratios of hospitalized and non-hospitalized notified leptospirosis cases in New Zealand, 1999 to 2017.

	Hospitalization Rates Ratio (95% CI)		
Serovars	All Cases	European	Māori
Ballum	1.70 (1.29–2.26) ***	1.81 (1.37–2.52) ***	0.75 (0.21–2.46)
Hardjo	0.80 (0.66–0.98) *	0.76 (0.60–0.96) *	1.33 (0.81–2.21)
Pomona	1.32 (1–1.74)	1.42 (1–2.03) *	1.52 (0.89–2.63)
Tarassovi	0.58 (0.37–0.89) *	0.68 (0.42–1.07)	0 (0–1.51)
Unknown serovar	1.26 (1.02–1.57) *	1.24 (0.97–1.59)	1.00 (0.59–1.70)
Total	1.16 (1.05–1.29) **	1.11 (0.98–1.26)	1.17 (0.89–1.56)

*** *p* < 0.001, ** *p* < 0.01, * *p* < 0.05.

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
