# Peer review of "Diverse Epidemiology of *Leptospira* Serovars Notified in New Zealand, 1999–2017"

_pathogens, 2020, doi:10.3390/pathogens9100841_

Round 1
Reviewer 1 Report
In the present work, the authors investigated the epidemiology of leptospirosis from 1999 to 2017 in New Zealand. The authors demonstrated that the 19-year average annual incidence is 2.01/100,000. Early (1999-2007) and late (2008-2017) study period comparisons showed a significant increase in notifications with serovar Ballum (IRR: 1.59, 24 95% CI: 1.22-2.09) in all cases and serovar Tarassovi (IRR: 1.75, 95% CI: 1.13-2.78) in Europeans and, a decrease in notifications with serovars Hardjo and Pomona in all cases. The stratification by occupation showed meat workers had the highest incidence of Hardjo (57.29/100,000) and Pomona (45.32/100,000), farmers had the highest incidence of Ballum (11.09/100,000) and dairy farmers had the highest incidence of Tarassovi (12.59/100,000). This is a good research article, and should be of interest to the readership of Pathogens. In my opinion, it is just need minor revision.
This study almost focus on New Zealand that is good. But if the authors can compare limilar data from other countries that may give a more general conclustion.
Author Response
Response to Reviewer 1 Comments
This study almost focus on New Zealand that is good. But if the authors can compare similar data from other countries that may give a more general conclusion.
Lines 307-318: A paragraph comparing epidemiological trends from other temperate countries has been added. It now reads:
The overall patterns of leptospirosis in New Zealand varied when compared to other temperate countries. Like New Zealand, improved diagnostics saw an increase in cases in Ireland with high-risk activities associated with occupations involving livestock [30]. However, unlike New Zealand, exposure in Ireland was also highly associated with recreational activities whereas exposure in Germany was as likely to be occupation as recreational [31] while in Italy exposure was more likely to be recreational than occupational [32]. Interesting, unlike New Zealand where the total monthly incidence of leptospirosis showed little change during the year, the seasonal pattern in European countries including Germany, Italy, France and Slovak Republic, showed high incidences between July to October [30-34]. While this observation is interesting, results from this study has shown that the total monthly incidence is not a good way to determine leptospirosis seasonality in New Zealand as each serovar has a different monthly pattern, likely due to different exposures and activities e.g. Hardjo and Pomona in meat workers and Tarassovi in dairy farmers.
Reviewer 2 Report
This is a stellar analysis of secondary data, and the writing is precise, concise and clear. The authors use their detailed knowledge and familiarity with the region to contextualise their findings exceptionally well. (An excellent use of lock-down?!!)
- Loess smoothing is mentioned extensively throughout the results. I think the methods section just needs a brief description and reference.
- The authors give a good interpretation of the results in the discussion and mention potential biases. In my opinion, the manuscript would be even better if there was a single short paragraph highlighting the main potential systematic biases of these secondary data that may impact on the key findings (particularly temporal trends and apparent association with ethnicity). A couple of important potential biases seem to be: (1) changes in the diagnostic tests used (this is mentioned but not in detail considering the long time period and the complex case definition); and (2) the identification of the population at risk using secondary sources (if the population at risk was under-estimated, then the rates by occupation could be over-estimated – I am particularly wondering if there are seasonal/temporary workers or family members who occasionally help out on farms that not included in these data, and that this could affect some ethnicities more than others?).
Some minor points:
- Line 63. Systematic
- Figure 3A legend. I suggest explaining the extents of boxes, lines and dots.
- Lines 188 and 190. Occupations.
- Line 215 and throughout manuscript. I suggest writing District Health Boards instead of DHB, throughout.
- Line 218 “for all cases” – this clause is a bit confusing, can it be deleted?
- Line 368. I suggested replacing ESR with Institute of Environmental Science & Research Ltd
- Line 393. Please make sure all R packages used are referenced.
Author Response
Response to Reviewer 2 Comments
1. Loess smoothing is mentioned extensively throughout the results. I think the methods section just needs a brief description and reference.
Line 400:401: a description of loess smoothing has been added. It now reads:
Temporal trends were visualized by report year (1999 to 2017) and report month (January to December) with locally weighted scatterplot smoothing (LOESS), a linear regression analysis tool that creates a smooth line through time plots to foresee trends.
Lines 422-425: References of analysis software and packages used have been added.
2. The authors give a good interpretation of the results in the discussion and mention potential biases. In my opinion, the manuscript would be even better if there was a single short paragraph highlighting the main potential systematic biases of these secondary data that may impact on the key findings (particularly temporal trends and apparent association with ethnicity). A couple of important potential biases seem to be: (1) changes in the diagnostic tests used (this is mentioned but not in detail considering the long time period and the complex case definition); and (2) the identification of the population at risk using secondary sources (if the population at risk was under-estimated, then the rates by occupation could be over-estimated – I am particularly wondering if there are seasonal/temporary workers or family members who occasionally help out on farms that not included in these data, and that this could affect some ethnicities more than others?).
Lines 319-330: A paragraph discussing biases have been added. It now reads:
It is important to note that while this study described total and serovar-specific trends, there are potential biases with this analysis. For example, the change in diagnostics can lead to misclassification of cases. If a region with a high incidence of Hardjo or Pomona switched from serological to PCR testing, this analysis will identify it as a decrease in Hardjo and Pomona cases in the late study period since the PCR positive Hardjo and Pomona cases will be classified as unknown serovar. In addition, the occupational incidence of farmers may be over-estimated because the population at risk denominator used the employed census of usually resident population. This denominator would not include migrant/seasonal/temporary workers and family/friends who may help at the farm but who may not have a farming occupation. Lastly, there are differences in trends seen in Europeans and Māori that may be attributed to physiological response to disease, lifestyle, health-seeking behaviors and changes in the at-risk population over time e.g. the number of meat workers decreased by 23% from the early to the late study period.
Some minor points:
3. Line 63. Systematic
Lines 63: Error edited as requested (systemic changed to systematic)
4. Figure 3A legend. I suggest explaining the extents of boxes, lines and dots.
Lines 152-157: Figure 3 legend has been edited as suggested. It now reads:
Figure 3. Age and sex of notified leptospirosis cases in New Zealand, 1999 to 2017. A) Box plot, B) frequency histogram, C) box plots stratified by serovar and D) frequency histograms stratified by serovar. Box plot widths are proportional to the square root of the number of cases, box plot lengths are displaying the interquartile range of the ages, the whiskers of the box are the minimum and maximum quartile of the ages, the line on the box marks the median ages and the dots represents outliers.
5. Lines 188 and 190. Occupations.
Lines 191 and 193: Error edited as requested (occupation changed to occupations)
6. Line 215 and throughout manuscript. I suggest writing District Health Boards instead of DHB, throughout.
DHB edited to District Health Board(s) in lines 218, 234, 396, 413-416, 418
7. Line 218 “for all cases” – this clause is a bit confusing, can it be deleted?
Line 218: Since we are comparing all cases with cases stratified by ethnicity, this cannot be deleted, however, I have rewritten it to avoid any confusion. It now reads:
For the remaining three serovars, Hardjo, Pomona and Tarassovi, the spatial patterns were similar for all cases and for cases stratified by ethnicity i.e. Hardjo predominated in Hawke’s Bay, Pomona predominated in Hawke’s Bay and Tairawhiti, and Tarassovi predominated in Waikato, Taranaki and Northland (Figure 6).
8. Line 368. I suggested replacing ESR with Institute of Environmental Science & Research Ltd
Line 396: ESR replaced with Institute of Environmental Science & Research Ltd as suggested
9. Line 393. Please make sure all R packages used are referenced.
Lines 421-424: edited with all packages and their references
Reviewer 3 Report
The article by S. Nisa entitled “Diverse epidemiology of Leptospira serovars notified in New Zealand, 1999-2017” describes the incidence as well as the demographics, temporal and spatial distribution of human leptospirosis cases in New Zealand.
It is a complete study, and I think it deserves to be published. Also, it shows the epidemiological heterogeneity of the different Leptospira serovars and suggests that this heterogeneity should be taken into account in prevention strategies.
Nonetheless, the authors should answer a few questions:
1.-How many of the included cases were confirmed cases, and how many were probable? This topic should be better described in the results.
2.-The authors must justify the choice of the two periods: early (1999-2007) and late (2008-2017). Given Figure 1, it would seem more appropriate to subdivide the late period into two periods (from 2008 to 2014, and the other from 2015 to 2017).
Author Response
Response to Reviewer 3 Comments
1.-How many of the included cases were confirmed cases, and how many were probable? This topic should be better described in the results.
Line 92 of the results section states the number of confirmed (n=1520) and probable (n=107) cases. In addition, lines 368-376 describes in detail what entails a confirmed and a probable case. The line “Due to the small number of probable cases, cases were not stratified by confirmed and probable. All analysis was performed on total cases” is now added to the methods section (lines 379-380).
2.-The authors must justify the choice of the two periods: early (1999-2007) and late (2008-2017). Given Figure 1, it would seem more appropriate to subdivide the late period into two periods (from 2008 to 2014, and the other from 2015 to 2017).
Thank you for raising this point. Considerable thought was given on how best to divide the study period. Since the focus of the paper is on the serovars, the early and late periods were dictated by the trend lines of Fig 1B which shows a change in trajectory around 2008 for known serovars.